# Evaporate-casting of curvature gradient graphene superstructures for ultra-high strength structural materials

Bing Lu[1,2], Li Yu ●[3], Yajie Hu[1], Ying Wang[1], Fei Zhao ●[2], Yang Zhao ●[2] ✉,
Feng Liu ●[3] ✉, Huhu Cheng ●[1] & Liangti Qu ●[1] ✉

In contemporary manufacturing, the processing of structural materials plays a pivotal role in enabling the creation of robust, tailor-made, and precise components suitable for diverse industrial applications. Nonetheless, current material forming technologies face challenges due to internal stress and defects, resulting in a substantial decline in both mechanical properties and processing precision. We herein develop a processing strategy toward graphene superstructure with a curvature gradient, which allows us to fabricate robust structural materials with meticulously designed functional shapes. The structure consists of an arc-shaped assembly of graphene nanosheets positioned at co-axial curvature centers. During the dehydration-based evaporate-casting process, the assembly is tightened via capillary effect, inducing local bending. By precisely tuning the axis-center distance and tilt angle, we achieve accurate control over the shape of obtained structure. Notably, internal stress is harnessed to reinforce a designed mortise and tenon structure, resulting in a high joining strength of up to ~200 MPa. This innovative approach addresses the challenges faced by current material forming technologies and opens up more possibilities for the manufacturing of robust and precisely shaped components.

Structural material processing plays crucial roles in virtually all aspects of modern manufacturing and industry, enabling the fabrication of customized, complex, and precision parts that are durable and well-suited for various applications, spanning from electronics to aerospace and automotive industries[1–3]. However, current material forming technologies suffer from internal stress or defects arising from the relative displacement of material molecules (or atoms) during the processing, which lead to a noteworthy attenuation of the intrinsic mechanical properties and a deterioration of processing precision[4–6]. In line with the development of two-dimensional materials over the past decades, microstructures constructed from graphene and its derivatives as basic building blocks have showcased tremendous potential, demonstrating that the bottom-up self-assembly is an effective strategy to address the processing challenge[7,8]. Through the deliberate regulation of interactions between graphene sheets, such as π-π interaction and hydrogen bonding, one can achieve designed structures encompassing porous structure (Fig. 1a-i)[9,10], layered structure (Fig. 1a-ii)[11,12], or array structure (Fig. 1a-iii)[13,14]. To alter the assembly's configuration and attain desired structural materials, an external force is necessary, while the resulting internal stress can be dissipated by relative slip between sheets. Nevertheless, random interlayer slippage of nanosheets in the above three assemblies will lead to unpredictable changes, and thus defects, in the resultant structure.

[1]Department of Chemistry, Key Laboratory of Organic Optoelectronics & Molecular Engineering, Ministry of Education, Tsinghua University, Beijing 100084, PR China. [2]Key Laboratory of Cluster Science, Ministry of Education of China, Key Laboratory of Photoelectronic/Electrophotonic Conversion Materials, School of Chemistry and Chemical Engineering, Beijing Institute of Technology, Beijing 100081, PR China. [3]State Key Laboratory of Nonlinear Mechanics, Institute of Mechanics, Chinese Academy of Sciences, Beijing 100190, PR China. ✉e-mail: yzhao@bit.edu.cn; liufeng@imech.ac.cn; lqu@tsinghua.edu.cn

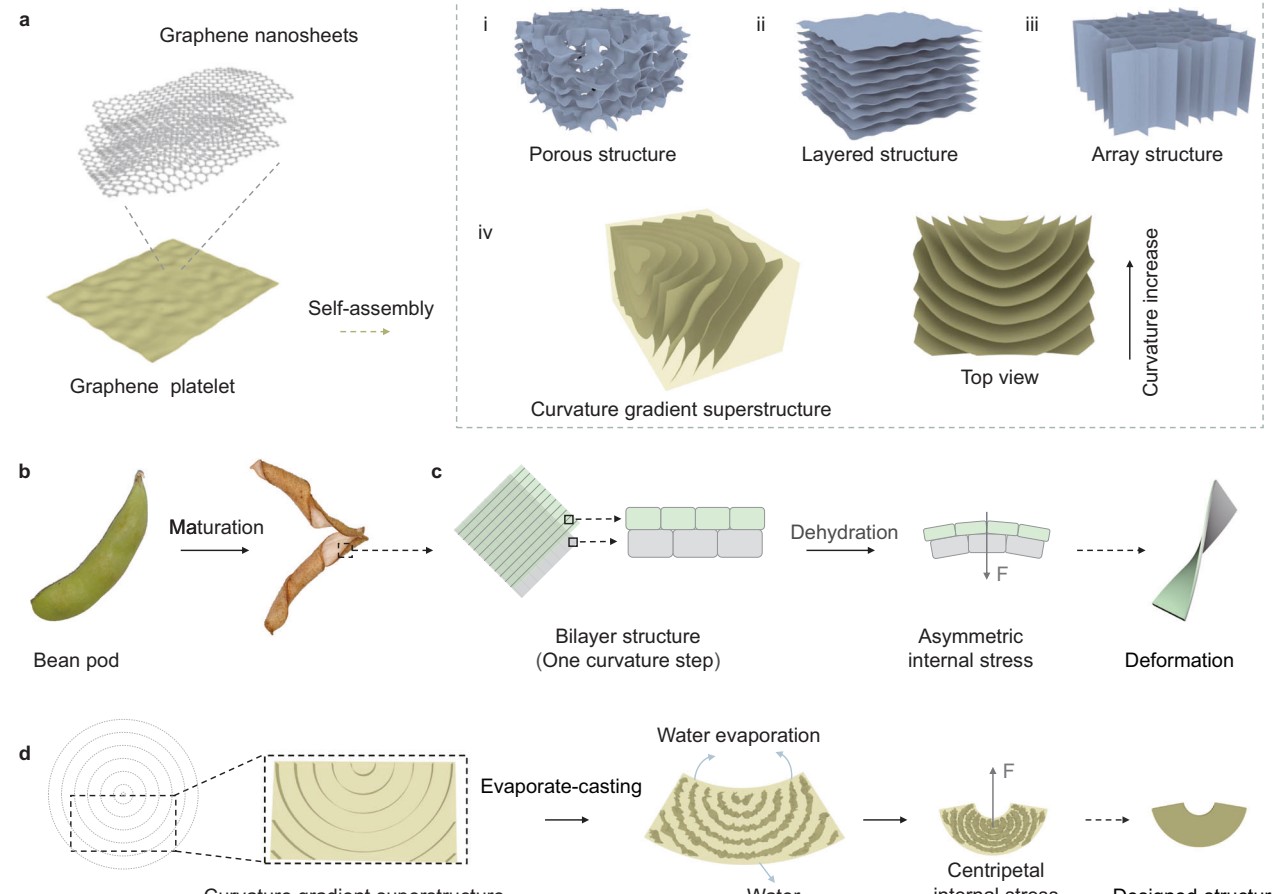

**Fig. 1 | Conceptual illustrations of evaporate-casting of *cg-G*. a** Graphene platelet derived from monolayer graphene nanosheets as basic building block for graphene-based functional architectures shown in i to iv: (i) porous structure, (ii) layered structure, (iii) array structure, and (iv) curvature gradient superstructure. The use of different colors (blue-gray and yellow) is to distinguish the common structure from our structure. **b** Deformation of the skin of bean pod. **c** Dehydration and deformation process of bilayer structure in the skin of bean pod. The two layers deem to be one curvature step. **d** Dehydration and evaporate-casting process of *cg-G*. The assembly is tightened by capillary action during water evaporating, and the curvature exhibits a gradient change, resulting in local bending driven by internal configurational force.

The gradient structures are introduced as a groundbreaking conceptual design, demonstrating that despite relatively low precision, the internal stress inherent in material forming can still be effectively harnessed to regulate the shapes of the resulting products. Owing to the well-organized structure along the gradient direction, including the gradient of functional groups (known as chemical gradient)[15] and gradient of pores (i.e., physical gradient)[16], stress generated during material forming induces differential deformation at various positions within the material, achieving the shaping. Although the gradient structures are innovative in dissipating internal stress for high robustness, it still constitutes an emerging technology, presenting challenges in utilizing internal stress as a driving force for shaping process.

We demonstrate here the fabrication of a graphene superstructure with curvature gradient (*cg-G*), and present a processing strategy to construct strong structural material with designed functional shapes. The structure comprises a circular arc-shaped assembly of graphene nanosheets positioned at co-axial curvature centers (Fig. 1a-iv). Our inspiration draws from the curvature junctions observed within the layered cells of a bean pod (Fig. 1b)[17–19]. The curvature arises as a result of the asymmetrical shrinkage of two layers during the dehydration process, ultimately inducing internal stresses that enhance the structural strength of the entire assembly while also driving deformation (Fig. 1c)[20,21]. Water molecules act as molecular-level plasticizers, engaging with the oxygen-containing

functional groups on the graphene nanosheets through hydrogen bonding, thus imparting mobility to the graphene nanosheets. The assembly is tightened by capillary action during the dehydration-based evaporate-casting process, and the curvature also exhibits a gradient change, consequently inducing local bending driven by internal stress (Fig. 1d). Through controllable coordination of axis-center distance and tilt angle, we can exert accurate control over the shape of the resulting casting. Remarkably, in designed mortise and tenon structure, internal stress can be leveraged to reinforce the formation, leading to the attainment of a super high-strength connection (~200 MPa) upon a specific strength of 143 MPa cm³ g⁻¹, surpassing that of metallic welding connection, or even integrally formed alloys. The entire process is characterized by its high level of controllability, feasibility, cost-effectiveness, versatility, and environmentally friendly attributes.

## Results

### Fabrication and characterization of *cg-G*

*cg-G* is produced through an alkaline-assisted hydrothermal synthesis using graphene oxide (GO) as precursor (see Methods for details, Supplementary Figs. 1 and 2). The graphene walls are aligned around the geometric axis, which concurrently serves as the axis of curvature of the obtained structure. Using laser processing technology, one can shape the materials with exceptional precision, achieving accuracy of ca. 10 micrometers when combined with an evaporate-casting process

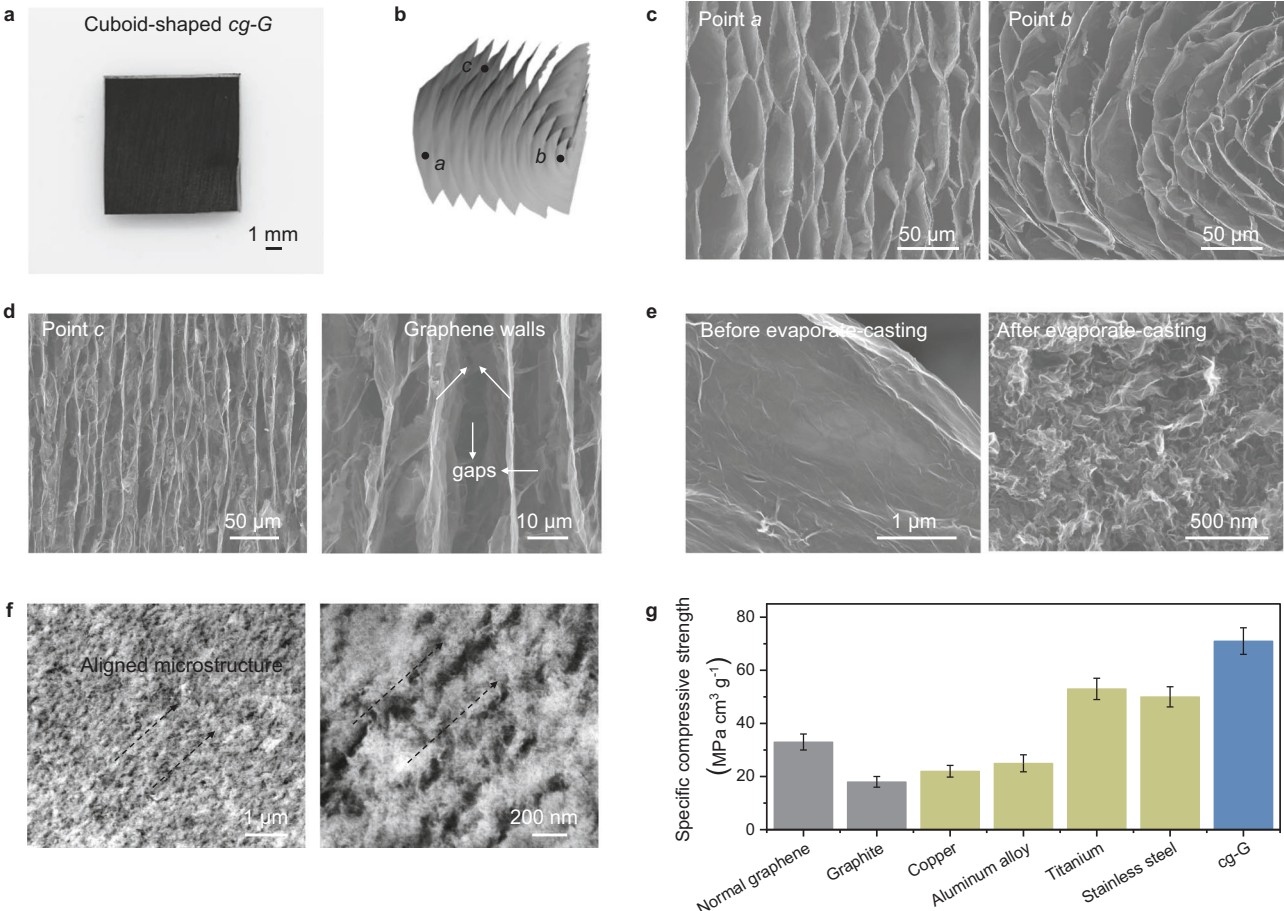

**Fig. 2 | Morphology and structure characterizations of *cg-G*. a** Photograph of a cuboid-shaped *cg-G*. **b** Schematic structure of *cg-G*. Three colorful points a (green), b (yellow), and c (purple) represent three different positions on the assembly. **c** SEM images of point a and point b in (**b**). The position close to the center of curvature shows obvious curvature gradient structure. **d** SEM image of point c and corresponding enlarged image. The structure of point c is composed of parallel graphene walls and gaps between them. **e** SEM images of graphene sheets in *cg-G* before and after evaporate-casting. **f** STEM images of point c in Fig. 2b after evaporate-casting. Yellow arrows represent the aligned direction of compact graphene walls. **g** Specific compressive strength comparison of various densified materials including normal graphene hydrogel after dehydration, graphite, kinds of metals, and *cg-G* after evaporate-casting (*n* = 3, error bar: standard deviation). Source data are provided as a source data file.

(Fig. 2a and Supplementary Fig. 3). The resultant superstructure is featured with a curvature gradient (Fig. 2b), where the curvature is smaller further from the curvature center and increases as one approaches the center. The scanning electron microscopy (SEM) investigations demonstrate that the structure at point *a* (farther from the curvature center) shows a nearly align parallel to the central axis of curvature (Fig. 2c), while the one at point *b* (near to the curvature center) exhibits neat remarkable curved arrays toward the concentric axis. The side view cross-section at point *c* (parallel to the curvature central axis) shows a similar parallel structure to the one at point *a* (Fig. 2d).

In a swelling state, the gaps amidst the walls become saturated with water molecules. The evaporate-casting process generates capillary forces through water evaporation, drawing the flat graphene assembly and yielding the protruding structures (Fig. 2e, Supplementary Figs. 4 and 5). Analysis via X-ray photoelectron spectroscopy indicates an oxygen composition of approximately 20% within the *cg-G*, affirming the presence of hydrophilic oxygen functional groups (Supplementary Fig. 6a). This observation emphasizes a robust interaction between water and the internal surfaces, thereby ensuring the potency of the capillary forces. Both Fourier transform infrared spectroscopy and energy dispersive spectroscopy before and after water evaporation indicate that the casting process does not alter the composition or content of functional groups on graphene sheets

(Supplementary Fig. 6b, d, e). Additionally, X-ray diffraction patterns demonstrate no discernible peak shift between the dehydrated and hydrated states (Supplementary Fig. 6c), signifying a consistent interlayer structure connecting the two graphene nanosheets within the assemblies. This suggests that the protruding structures retain the intrinsic mechanical attributes of the graphene assembly. Upon complete dehydration, an aligned microstructure is observed under scanning transmission electron microscopy (STEM), wherein the protrusions are mechanically interlocked in a serrated configuration (Fig. 2f, Supplementary Fig. 7). Due to this distinctive structure, *cg-G* after deformation demonstrates a density of ~1.5 g cm$^{-3}$ and high compressive strength of up to 70 MPa/(g/cm³), surpassing that of various metals including stainless steel (SUS304) and doubling the strength of traditional graphene-based structural materials (Fig. 2g, Supplementary Fig. 8).

## Controllable shaping based on curvature gradient

To achieve precise control over curvature gradients, we utilized the distance (*d*) extending from the geometric center (O) of the samples to the curvature axis of the superstructure (O') as a governing parameter (Fig. 3a, 3b-i). In situations where the curvature axis of the graphene superstructure is perpendicular to the plane of the cross-shaped samples, variations in the distribution of curvature gradients within the sample result in diverse bending behaviors during the evaporate-

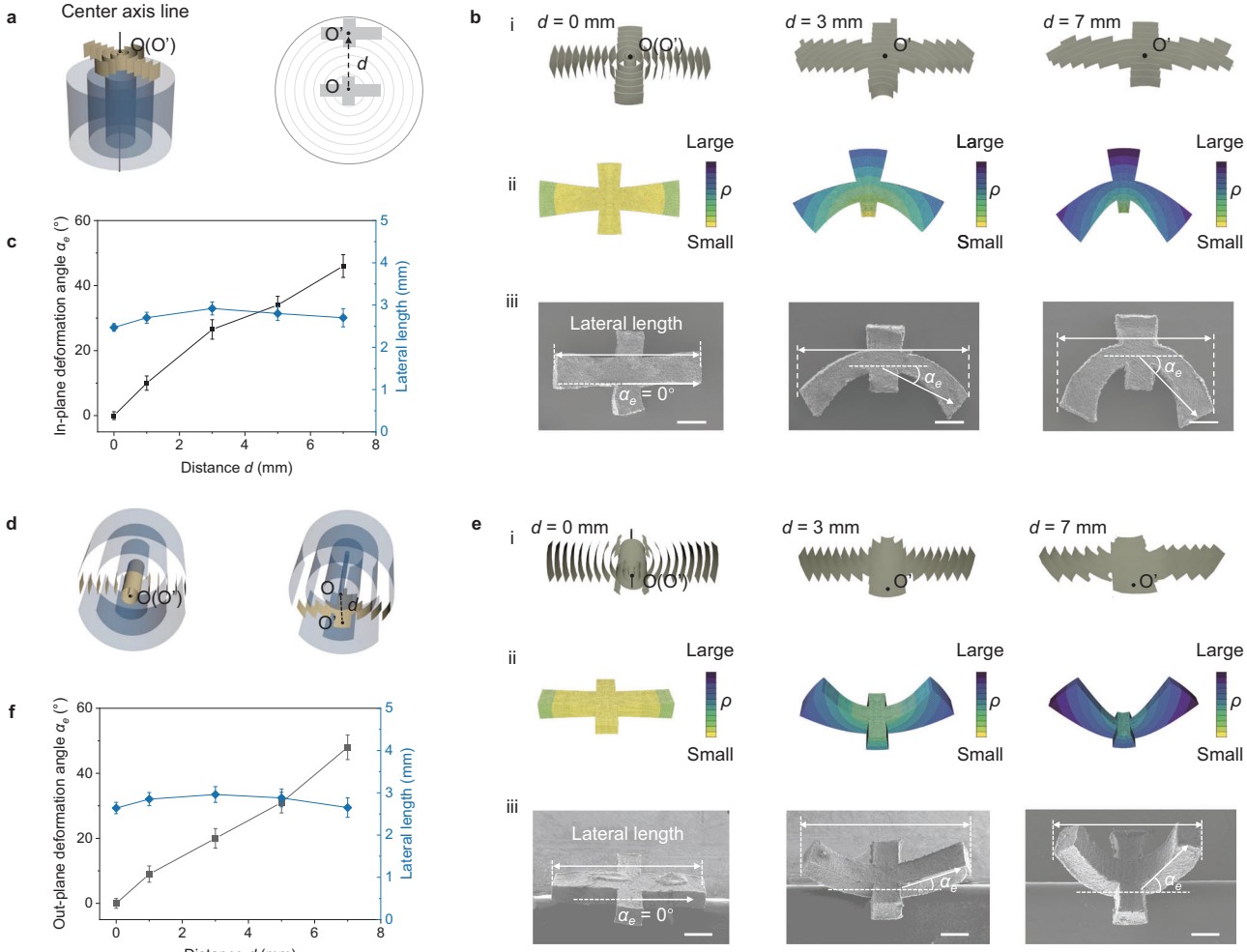

**Fig. 3 | In-plane and out-plane evaporate-casting processing of *cg-G*. a–c** In-plane processing. **a** Schematic diagrams of cross-shaped *cg-G* with an axis-center distance (*d*) perpendicular to centric axis of curvature. **b** Demonstrations of cross-shaped *cg-G*s with different microstructures before and after deformation: (i) Schematic diagrams of three cross-shaped *cg-G* with different *d* values (*d* = 0, 3, 7 mm) on in-plane direction. (ii) Gradient distribution of the curvatures and theoretical predictions of the deformation of the three *cg-G*s based on finite element analysis (FEA). (iii) SEM images of the three deformed *cg-G*s. The colors in FEA from yellow to dark blue indicate an increase of the radius of curvature ($\rho$). The degree of deformation can be represented by the included angle ($\alpha_e$). Scale bar: 500 µm. **c** $\alpha_e$ and lateral length of final graphene architectures as a function of *d* (*n* = 3, error bar: standard deviation). **d–f** Out-plane processing. **d** Schematic diagrams of *cg-G*s with different *d* values along the centric axis of curvature. **e** Demonstrations of cross-shaped *cg-G*s with different microstructures before and after deformation: (i) Schematic diagrams of three cross-shaped *cg-G*s with different *d* values (*d* = 0, 3, 7 mm) on out-plane direction. (ii) Corresponding gradient distribution of the curvatures and theoretical predictions of the deformation of the three *cg-G*s based on FEA. (iii) SEM images of the three deformed *cg-G*s. The degree of out-plane deformation can be represented by $\alpha_e$. Scale bar: 500 µm. **f** $\alpha_e$ and lateral length of final graphene architectures as a function of *d* (*n* = 3, error bar: standard deviation). Source data are provided as a source data file.

casting process (Fig. 3b-ii and 3b-iii, Supplementary Figs. 9 and 10, Supplementary Video 1). The deformation of samples with diverse curvature gradients can be anticipated via computational prediction (see Methods for details). While the level of shrinkage is notably high, the overall form of the cross-shaped samples is upheld when *d* = 0. This phenomenon arises due to the counteraction of configurational forces in these sections, which is a consequence of the symmetric microstructures aligning along the in-plane direction. In contrast, the prominence of the differential distribution of the shrinkage coefficient amplifies as *d* increases (Supplementary Fig. 11). In this context, the degree of shrinkage escalates alongside a more substantial curvature gradient. As a result, these cross-shaped *cg-G*s exhibit in-plane deformations toward the curvature center direction following the process of evaporate-casting.

The microscopic observation via SEM (Fig. 3b-iii) experimentally validated the correlation between deflection and curvature gradient. Samples with a carefully designed curvature gradient exhibited corresponding bending angles ($\alpha_e$) along the longer sides (the long side of the initial cross-shaped *cg-G*) that aligned with the values derived from computational predictions ($\alpha_c$) (Fig. 3b-ii, Supplementary Fig. 9b). This alignment suggests a high level of uniformity in the self-deformation facilitated by the curvature gradient. The lateral dimension, which signifies the extent of the long side of the initial cross-shaped *cg-G* after deformation, is utilized to exemplify the pronounced regularity of self-deformation. Notably, despite the progressive augmentation in the curvature gradient, the lateral length remains invariant (Fig. 3c). The alterations in shape along the horizontal direction are relies on the decreasing gap width of the wall structures. As a result, the constant lateral dimension not only denotes a consistent curvature gradient but also a contraction that closely approximates the characteristics of an ideal coaxial circular arc model (see Supplementary Fig. 12 for details), which concurs with the structural characterization elucidated in Fig. 2. More examples of the in-plane deformation are shown in Supplementary Figs. 13 and 14.

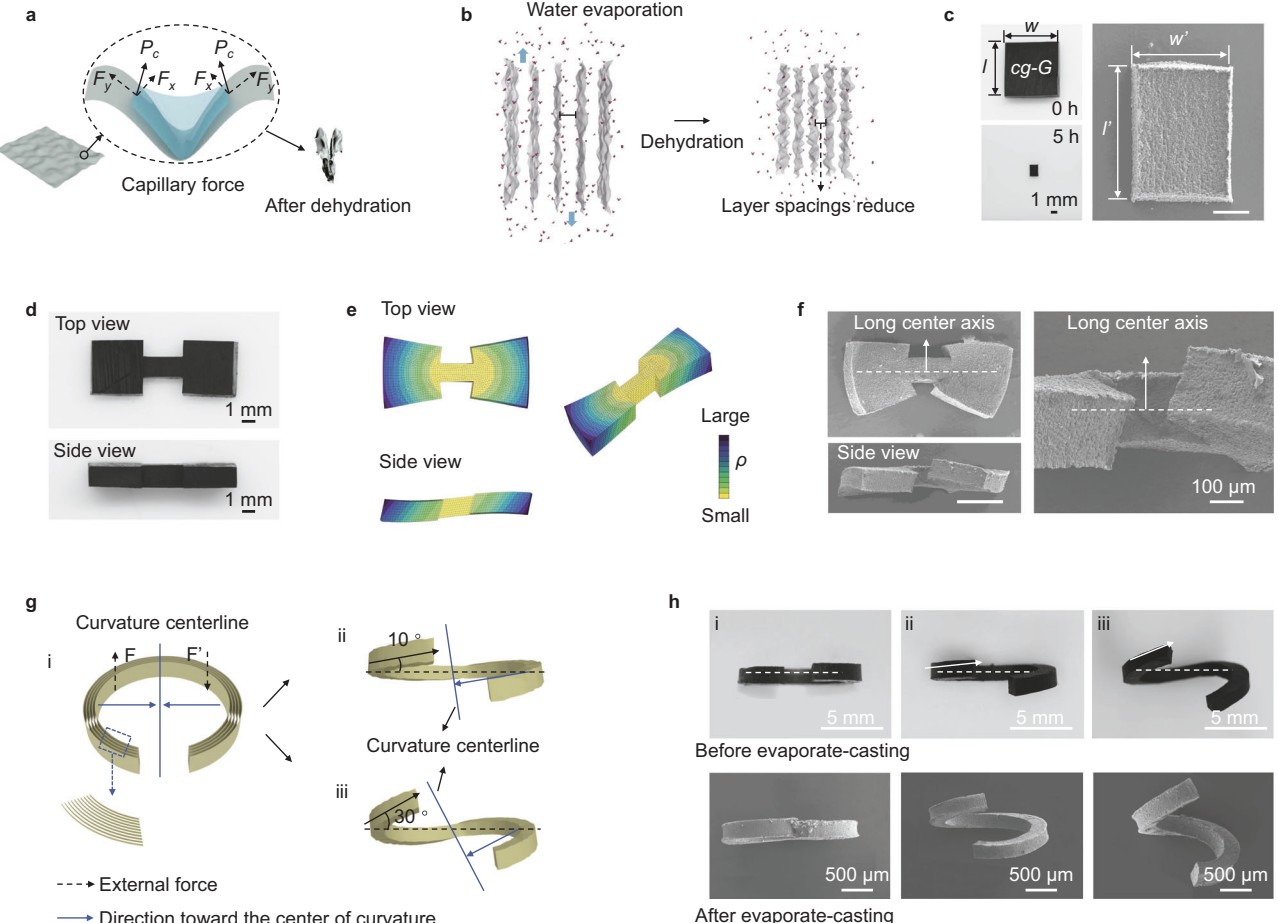

**Fig. 4 | Mechanism of evaporate-casting of *cg-G*. a** Schematic of capillary force ($P_c$) of water in the groove of a monolayer graphene sheet, and the deformation of graphene groove after dehydration. The capillary force acts perpendicular to the water surface and can be divided into two force modes: one is normal to the graphene sheet ($F_x$) and the other is along the sheet ($F_y$). **b** Schematic of the dehydration process of graphene walls arrayed in parallel during evaporate-casting. **c** Photographs of cuboid-shaped *cg-G* comprising a centrosymmetric curvature gradient structure after dehydration of 0 h and 5 h, and the corresponding SEM image of dehydration for 5 h. Scale bar: 500 μm. Initial size: 8 mm × 8 mm × 2 mm. The ratios of the two sides along and perpendicular to the wall plane before ($l, w$) and after ($l', w'$) dehydration are different, specifically, $w/w' > l/l'$. **d** Digital images of top and side views for dumbbell-shaped *cg-G*. **e** Gradient distribution of the

curvatures and theoretical predictions of the deformation of the dumbbell-shaped *cg-G*. **f** SEM images of the dumbbell-shaped *cg-G* after evaporate-casting. The colors in FEA from yellow to dark blue indicate an increase of the radius of curvature ($\rho$). Scale bar for the two images on the left: 500 μm. **g** Schematic diagrams of the microstructure of (i) arc-shaped *cg-G*, and corresponding schematic diagrams of the assembly after being twisted to different angles (small angle of 10° (ii) and large angle of 30° (iii)). F and F' represent two forces with equal magnitude but in opposite directions. After being twisted, the direction towards the curvature center (blue arrow) shifts from in-plane to out-plane. **h** Digital photos and SEM images of three *cg-G* assemblies with a twisted angle of 0° (i), 10° (ii), and 30° (iii) before and after evaporate-casting.

We also examined a scenario where the axis of curvature is parallel to the plane of the cross-shape specimen, and we denoted the distance between the geometric center point and the curvature axis as $d$ (Fig. 3d, 3e-i). The deformation induced by the evaporate-casting process was simulated computationally (Fig. 3e-ii, Supplementary Fig. 15a). Despite the significant contraction (high shrinkage coefficient level), the fundamental configuration of the cross-shaped samples remains unaltered when $d = 0$. This phenomenon can be ascribed to the equilibrium between internal stresses within symmetric micro-structures. The distribution of the shrinkage coefficient becomes more pronounced as the curvature gradient increases (Fig. 3e-iii), signifying that the relationship between the degree of out-plane bending and curvature gradient mirrors that of in-plane deformation. Additionally, a distinct out-plane deformation becomes evident, aligning with the computational predictions. The independence of the lateral length on the curvature gradient underscores the nearly ideal arrangement of co-axial circular arc-shaped graphene assemblies, reinforcing the con-cordance between observations and expectations (Fig. 3f, Supple-mentary Fig. 15b). The consistency of these phenomena with the

results of in-plane deformation experiments highlights satisfactory controllability facilitated by the curvature gradient in self-shaping processes.

## Mechanism of self-shaping

The evaporate-casting enabled self-shaping relies on the hierarchical deformation of graphene walls. Water molecules serve as plasticizers, enabling relative sliding of graphene nanosheets within the walls. In this context, as water within surface wrinkles evaporates, capillary forces prompt each graphene nanosheet to crumple (Fig. 4a). Conse-quently, protrusions emerge on the dehydrated wall surfaces (Fig. 2e, f), causing shrinkage along the plane of the wall. It's worth noting that due to robust π-π interaction forces between graphene sheets, the crum-pled walls retain their form even when in contact with water. On an assembly scale, capillary forces exert a tautening effect on the wall structures, diminishing gap sizes and hence yielding a compact arrangement (Fig. 4b). The emergence of protrusions and the con-striction of wall structures collectively bring about macroscopic shape contraction in orthogonal directions. This behavior can be reflected on

a cuboid sample comprising a centrosymmetric curvature gradient microstructure (Fig. 4c, Supplementary Figs. 16 and 17). The ratio of contraction along the wall plane ($l'$) compared to that perpendicular to the wall plane ($w'$) was measured at 1.4, indicating the anisotropic contraction behavior that could significantly bolster the efficacy of the curvature gradient induced deformation. This casting method demonstrates high stability. The ratio of $l'/w'$ is determined by the microstructure of $cg$-$G$ (Supplementary Fig. 18), and remains unaffected by the surrounding environment during water evaporation (Supplementary Fig. 19).

Then we delve into how the self-tightening effect of $cg$-$G$s during the evaporate-casting process is harnessed to programmatically transform simple profiles into intricate structures. According to the Kelvin curvature effect[22], water molecules are more likely to escape from curved surfaces. This observation aligns with our experimental findings that water evaporates more rapidly from the side with larger curvature. With water evaporating, moisture near highly curved graphene sheets becomes depleted first, leading to more pronounced deformations compared to regions with lower curvature. Considering these mechanisms, we can postulate the following relationship between radial strain ($\varepsilon_r$, strain towards the center of the curvature) and curvature radius ($\rho$) of $cg$-$G$ during evaporate-casting:

$$\varepsilon_r = \varepsilon_0 + bf(\rho)(\varepsilon_r < 0) \qquad (1)$$

where $\varepsilon_0$ (<0) represents the uniform shrinkage strain (the curvature radius equals zero). $b$ (>0) is the curvature deformation coefficient, indicating the sensitivity of deformation to changes in curvature radius. $f(\rho)$ is a function of $\rho$, signifying that as the curvature radius increases, the shrinkage deformation decreases. In this study, when analyzing $cg$-$G$ slices consists of graphene walls positioned at co-axial curvature centers (Supplementary Fig. 20), we assumed a relationship between $\varepsilon_r$ and $\rho$, i.e., $\varepsilon_r = \varepsilon_0 + b\rho^2$ (see Supplementary text for details). Specifically, the curvature microstructures of $cg$-$G$ slices can be described by their $d$ and $\theta$ values. Based on above relationship, we investigated the deformation of $cg$-$G$ with different $d$ and $\theta$, and provided corresponding analytical expressions (see Supplementary text for details). Theoretical results not only effectively accord with the deformation states discussed in Fig. 3, but also accurately predict deformations of dumbbell-shaped $cg$-$G$s (see Figs. 21–23 for a detailed discussion). For example, the FEA results and SEM images reveal that the dumbbell architecture (Fig. 4d, with a $\theta$ value of 45°) presents twists around the long center axis after deformation (Fig. 4e, f), resulting in a stereo and centrosymmetric final geometry, and bilateral cuboids transformed to be approximate trapezoidal cube due to the in-plane deformation. These results validate the mechanism of curvature gradient-induced bending deformation.

Apart from the self-shaping $cg$-$G$ structures that can be predicted and understood through having defined the relevant $\theta$ and $d$ parameters, this behavior can be further programmed through external force which allows the assembly to exhibit self-plastic capability like metals. To demonstrate this point, we applied two forces of equal magnitude and opposite direction on both ends of an arc-shaped $cg$-$G$ (Fig. 4g), resulting in the initial twist deformations with different bending angles, such as a small angle of 10° (Fig. 4g-ii) and a large angle of 30° (Fig. 4g-iii), respectively. After deformation, the direction towards the curvature center (blue arrow) shifts from in-plane to out-plane. The coarse-grained molecular dynamics (CG-MD) calculation model show that the energy of the system initially increases and then decreases as raise of twist angles because of inter-adhered graphene walls driven by strong π−π interactions (Supplementary Fig. 24). As the degree of deformation intensifies, the quantity of adhering walls escalates, causing a continual reduction in the system's energy. When the energy drops below zero, the system achieves stability, and the

structure displays plasticity, enabling it to preserve the shape shaped by external forces (Fig. 4g-ii, iii).

The $cg$-$G$ slices can be shaped due to their distinctive curvature gradient structures through evaporate-casting. As water evaporates, all arc-shaped $cg$-$G$ architectures are expected to undergo alterations in their initial profiles (Fig. 4h). The one with symmetrical curvature center structure transformed into a closed-loop structure during the evaporate-casting process (Fig. 4h-i), aligning with the previously mentioned self-shaping principles. When subjected to additional forces with initial twist angles of 10° (Fig. 4h-ii) and 30° (Fig. 4h-iii), the direction pointing towards the center of curvature changed, signifying the existence of out-of-plane components of the asymmetry within the curvature structure. The changes directly resulted further structural deformations out of plane. In SEM images, two helix-shaped $cg$-$G$ architectures with the same deformation mode but different pitches can be observed, and the twist angle is larger than it before deformation. These deformations resulted from the configurational forces generated by the combined effects of the initial curvature gradient and the additional twist angle during dehydration. These results confirm the previously established deformation mechanism.

### Enhanced mechanical strength through hierarchical self-tightening

We constructed a gasket structure (Fig. 5a), aligning the curvature center axis with the geometric center axis, to investigate the self-tightening effect induced by evaporate-casting across a highly symmetric curvature gradient. Impressively, when five micro-fixators were placed between two thick stainless steel plates (Fig. 5b), the assembly could withstand nearly one ton of weight as exemplified by a car (weighing ~1200 kg) running over on the top (Fig. 5c). Throughout the compression process, the gasket structure remained unchanged (Supplementary Fig. 25), while distinct indentations appeared on the stainless steel plate (Fig. 5d), underscoring the superior strength of the gasket structure in comparison to stainless steel. By taking virtue from self-deformation behaviors induced by curvature gradient structures, we showed that various preset architectures could be accurately constructed, such as a blooming flower-like configuration (Supplementary Fig. 26a–c). Additionally, the macroscopic-scale self-tightening is demonstrated by the mortise and tenon structure. We employed asymmetric shrinkage to craft samples featuring mortise structures (Fig. 5e). For the tenon structure, we positioned the curvature center axis at the center of the mortise circle. This arrangement enabled the tenon structure to display an inward deformation tendency during the coupling process with evaporate-casting, ultimately achieving a macroscopic self-tightening effect. Upon examining the microscopic morphology of the joint, we observed slight deformation in the shoulder part of the mortise structure (Supplementary Fig. 26d), indicating the centripetal deformation inherent in the tenon structure. A tight juncture of the mortise and tenon structures revealed by the SEM investigation further confirmed the self-locking behavior (Fig. 5f, Supplementary Fig. 26e). Remarkably, a self-tightening mortise and tenon configuration, measuring merely 1 cubic millimeter in volume, demonstrated the capacity to lift a weight weighing 300,000 times its own mass (Fig. 5g, Supplementary Video 2). Its specific tensile strength reached up to 143 MPa cm³ g⁻¹ (Fig. 5h, Supplementary Figs. 27 and 28), comparable to welded metals[23,24], and far superior to various types of connectors including self-healing polymer[25–27], and 3D printed structures[28–30] (Supplementary Table 1). Energy utilization can also be calculated through the thermal transfer power during the material preparation process, and the mechanically interlocked structure exhibits values that are 1 to 4 orders of magnitude higher than those of the aforementioned contrast materials (the calculation of energy utilization is detailed in Supplementary Information). Concurrently, its substantial materials and size expandability, and adaptability serve to

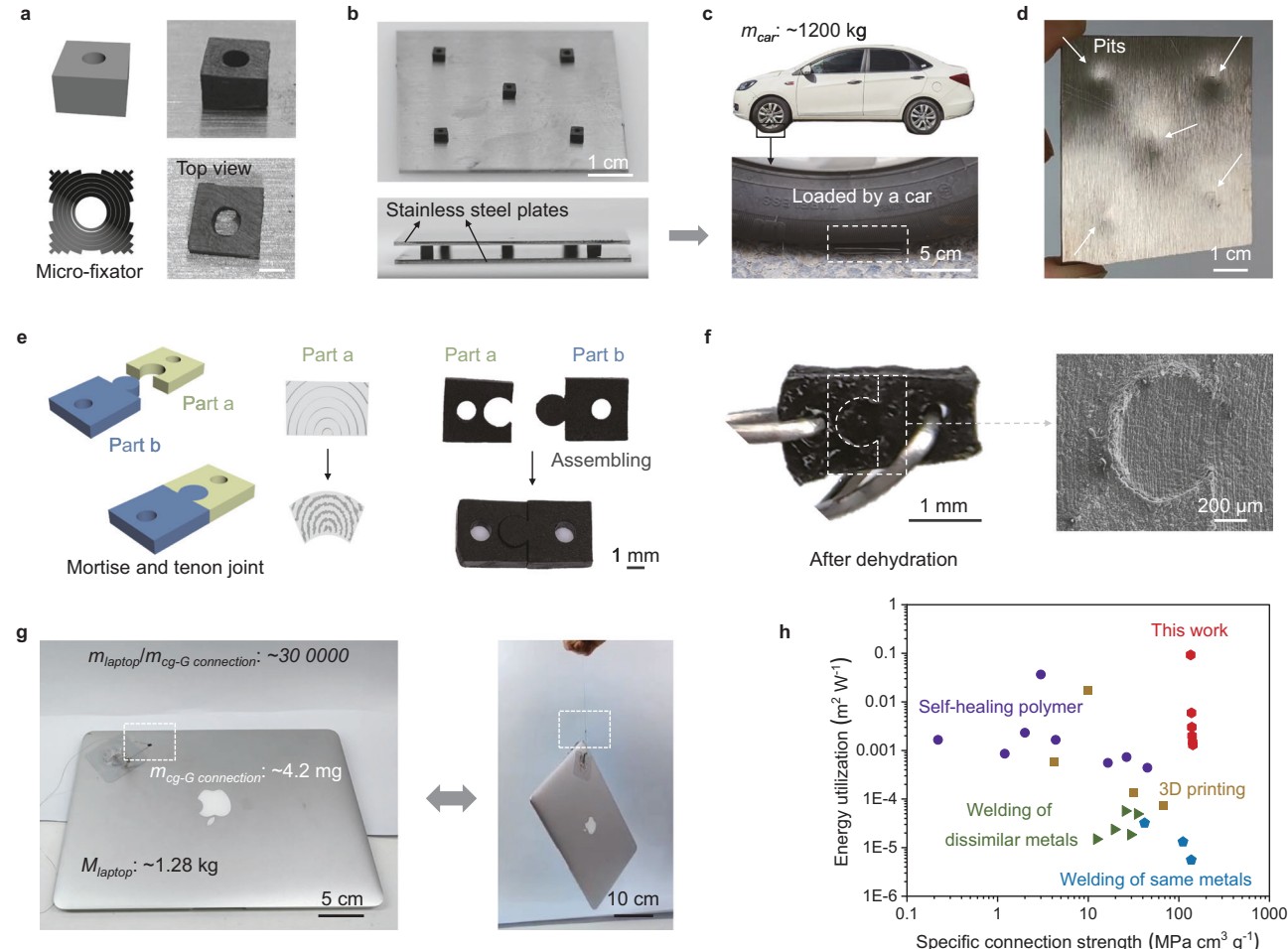

**Fig. 5 | Mechanical strength and demonstrations of *cg-G*s after evaporate-casting. a** Schematic and optical images of cuboid-shaped *cg-G* with circular void in the middle after evaporate-casting. Scale bar: 1 mm. **b** Optical images of five cuboid-shaped *cg-G*s with circular void in the middle after evaporate-casting. Those cuboids are put on a stainless steel plate and sandwiched by two pieces of stainless steel plates without binder (thickness: 1 mm). **c** Optical images of a car upon the five graphene architectures sandwiched by two pieces of stainless steel plates and the enlarged image. **d** Five obvious pits impressed on steel plate after being run over by car. **e** Schematic and corresponding optical photos of *cg-G* blocks with mechanical interlocking structure. **f** Optical and SEM images of the clasped structure after evaporate-casting process. **g** Mechanical performance display of clasped structure. A single interlocking structure could pull up a laptop (~1.28 kg). **h** An Ashby diagram showing the energy utilization versus specific connection strength of graphene assemblies with mortise and tenon structure, as well as various 3D printing structures, self-healing polymers, and some typical welded same and dissimilar metals. Source data are provided as a source data file.

position it as a promising, innovative, and eco-friendly technology for structural connections (Supplementary Figs. 29 and 30).

The evaporate-casting of curvature gradient graphene super-structures presented herein allow for the controllable fabrication of graphene micro-architectures with diverse designed functional shapes and high mechanical strength. With the controllable design of uniform arc-shaped assembly of graphene nanosheets positioned at co-axial curvature centers, configurational force pointed to the center of curvature generated during dehydration. As a result, the assembly is tightened during the dehydration-based evaporate-casting process via capillary force, inducing local bending. Through precisely tuning the axis-center distance and tilt angle, we realized accurate control over the shape of obtained evaporate-casting structure. In addition, mortise and tenon structure is induced to this work and the internal stress can be leveraged to reinforce the formation, leading to a high-strength connection (~200 MPa), surpassing that of multiple materials with connection structures, and even welded metals. Compared with tra-ditional welding technology, the proposed evaporate-casting proces-sing can occur spontaneously at room temperature, greatly reduced the threshold and energy consumption of the combination reaction. The entire process is characterized by its high level of controllability,

feasibility, cost-effectiveness, versatility, and environmentally friendly attributes.

# Methods
## Preparation of the GO dispersion
GO was prepared via the modified Hummers method[31]. Firstly, 10 g of graphite powder (mesh number: 200) and sodium nitrate (3 g) were added into a beaker (3 L) containing 260 mL of concentrated sulfur acid (98%), and stirred at 0 °C for 2 h. Then, 30 g of potassium per-manganate was carefully added under stirring, keep for 2 h. After that, the beaker was transferred to a constant temperature water bath of 35 °C and kept for stirring for 1 h. Thereafter, 460 mL of cold deionized water was dropwise added into the mixture and kept at 98 °C for 30 min. Additional 1 L of cold deionized water and 20 mL of hydrogen peroxide were poured into the solution. Finally, the product was centrifuged and washed by deionized water repeatedly until the pH changed from acidity to neutral, then the GO dispersion was produced.

## Fabrication of random graphene assembly
We prepared graphene hydrogel with random structure through hydrothermal process with GO as precursor. Specially, 60 ml GO

aqueous (3.5 mg mL$^{-1}$) dispersion was added into a 100 ml Teflon-lined autoclave, and treated by a hydrothermal process (180 °C for 6 h) in air oven, resulting in a normal cylindrical graphene hydrogel with random microstructures.

## Fabrication of *cg-G*

60 ml GO aqueous dispersion (3.5 mg mL$^{-1}$) was added into a 100 ml Teflon-lined autoclave and add certain amount of alkali (e.g., ammonium hydroxide (the volume ratio of GO aqueous dispersion to ammonium hydroxide (concentration: 32%) is 60:1), potassium hydroxide (0.15 mol L$^{-1}$)) under stirring. Keep stirring for 5 min and then treated by a hydrothermal process (180 °C for 6 h) in air oven, resulting in a cylindrical graphene hydrogel with ordered concentric circle microstructures. Subsequently, the prepared hydrogel was immersed in 1 M HCl solution for 6–8 h. After thoroughly rinsing to eliminate any lingering acid, the pure *cg-G* was obtained.

After adding alkali, GO sheets could transform from disorder state into long-range oriented lamellar phases. Benefiting from the reaction between alkali and carboxyl on GO sheets, the hydrogen bond interaction between GO sheets decreased distinctly and electrostatic repulsion increased (Supplementary Fig. 1). Under following stirring process, GO sheets tuned their direction vector in real time and maintained an overall trend of parallel to each other. The outline of fabricated hydrogel is determined by the inner surface of reaction still because of the surface anchoring effect, as well as the aligned arrangement of GO sheets. Changing the lining shape of Teflon-lined autoclave, graphene assemblies with different geometry were prepared because of the surface anchoring effect, as well as the aligned arrangement of GO sheets. By adjusting the concentration of GO aqueous dispersion (1.5 mg mL$^{-1}$, 2.5 mg mL$^{-1}$, 3.5 mg mL$^{-1}$, 4 mg mL$^{-1}$) and varying the quantity of alkali used, graphene hydrogels with different structures can be prepared (Supplementary Fig. 2).

*cg-G*s were fabricated via sculpturing graphene hydrogel with concentric circle microstructures by an ultraviolet laser cutting system (LAJAMIN 3D laser) conducted by computer operation system. By adjusting both the axis-center distance (*d*) and the tilt angle (*θ*: angle between the center axis of initial cylinder profile and the normal axis of target profile, changing from 0° to 90°), a diverse range of *cg-G* can be fabricated.

## Fabrication of complex *cg-G*

First, arc-shaped *cg-G* is fabricated via sculpturing graphene hydrogel with concentric circle microstructures (*θ* = 0°) by laser cutting system. Then, through the manipulation of one end of the arc structure while twisting the opposite end, external forces of equal magnitude but opposing directions can be simultaneously exerted at precise points on the target structure. By adjusting the twisting angle of the target structure, various complex *cg-G*s can be created.

## Fabrication of *cg-G* architectures via evaporating-casting process

Put as-prepared *cg-G* slices on a Teflon substrate with glazed surface. The surface of Teflon is super-hydrophobic, which can decrease the influence from the detect between substrate and hydrogel furthest. Standing the hydrogel in ambient atmosphere or convection oven (35 °C−65 °C) for a certain time, and *cg-G* architectures could be prepared via shrinking and deforming spontaneously.

## Characterization

The morphologies of the samples were investigated using a scanning electron microscope (SEM, SU-8010). HADDF-STEM images were obtained by FEI TecnaiTF20 transmission electron microscope. X-ray photoelectron spectroscopy was conducted on ULVAC-PHI PHI Quantro SXM X-ray photoelectron spectrometer. X-ray diffraction pattern was conducted on SmartLab multiple crystal X-ray

spectrometer. Mechanical tests were carried out using a Shimadzu AGS-X. Optical images were acquired on ZEISS Axio Scope. The graphene hydrogel assembly was sculptured by a 355 nm ultraviolet laser cutting system (Beijing Lagamin Laser Co., LM-UVY-5S-Y) under ambient conditions.

## Finite element analysis (FEA)

Due to the varying distances from the center, the curvature and the degree of contraction of the material differ at different locations. Considering the known micro-mechanisms, we have set up the following strain field in FEA, $\varepsilon(x, y, z) = \varepsilon_0 + b(x^2 + y^2) = \varepsilon_0 + br^2$, $\varepsilon = 0$ at the outermost region. Among them, $\varepsilon_0 < 0$ represents the strain at the center, and $b > 0$ the curvature deformation coefficient. The contraction deformation is maximum at the center. As moving toward the outer regions, the degree of contraction gradually decreases. It is worth noting that graphene is distributed in a ring shape within the *x-y* plane in the specimen, while it is approximately uniformly distributed along the *z*-direction. Therefore, uniform loading in the *z*-direction is set to reflect this characteristic. It is also noted that the change in water evaporation rate during the deformation of graphene hydrogels. Thus, the set strain rate decreases as the deformation increases.

All of the above-introduced deformation fields are achieved through an equivalent temperature field and implemented within the finite element solver ABAQUS by using the user subroutine to define incremental strains. The overall finite element model was meshed with 8-node brick (C3D8) elements.

## Coarse grain molecular dynamics (MD) simulation

Coarse grain molecular dynamics (MD) simulation is performed to uncover the deformation mechanism during shrinking process. In all models, each graphene sheet or platelet is depicted by coarse grain particles, which compose a square or rectangle sheet. According to Ref. [Steven Cranford], the total energy of graphene sheets or platelets could be written as

$$\phi_{tot} = \phi_T + \phi_\varphi + \phi_\theta + \phi_{ad,gp} + \phi_{ad,wp} \quad (2.1)$$

where the first three terms are harmonic potentials given by

$$\phi_T = \frac{1}{2}K_T(r - r_0)^2, \phi_\varphi = \frac{1}{2}K_\varphi(\varphi - \varphi_0)^2, \phi_\theta = \frac{1}{2}K_\theta(\theta - \theta_0)^2 \quad (2.2)$$

where $K_T = 1000$, $K_\varphi = 0.1$ and $K_\theta = 0.1$. The equilibrium distance and angles are $r_0 = 1$, $\varphi_0 = 180°$ and $\theta_0 = 90°$. Note that here the Lennard-Jones unit system are applied. Lennard-Jones potential is used to describe the adhesion interaction between graphene particles,

$$\phi_{ad,gp} = 4\varepsilon_{ad,gp}\left[\left(\frac{\sigma_{ad,gp}}{r}\right)^{12} - \left(\frac{\sigma_{ad,gp}}{r}\right)^6\right] \quad (2.3)$$

Where $\varepsilon_{ad,gp} = 1$, and $\sigma_{ad,gp} = 0.89$. The adhesion interaction between graphene particles and water particles,

$$\phi_{ad,wp} = 4\varepsilon_{ad,wp}\left[\left(\frac{\sigma_{ad,wp}}{r}\right)^{12} - \left(\frac{\sigma_{ad,wp}}{r}\right)^6\right] \quad (2.4)$$

where $\varepsilon_{ad,wp} = 3$, and $\sigma_{ad,wp} = 0.89$. Besides, LJ potential is also used to depict the interaction between water particles

$$\phi_{ad,wp} = 4\varepsilon_{wp}\left[\left(\frac{\sigma_{wp}}{r}\right)^{12} - \left(\frac{\sigma_{wp}}{r}\right)^6\right] \quad (2.5)$$

where $\varepsilon_{wp} = 1$, and $\sigma_{wp} = 0.955$. The periodic boundary condition is adopted. To simulate the shrink process, the simulation box size is gradually reduced with water particles decreasing accordingly.

In some calculation models, parameters are changed due to the simulation scenario and scale difference. To be specific, the calculation model in Supplementary Fig. 24 does not include water particles, and the parameters $K_T = 100$, $K_\varphi = 10$, $\sigma_{ad,gp} = 0.4$; in the calculation model (Supplementary Fig. 5), $K_T = 100$, $K_\varphi = 100$, and $\varepsilon_{ad,wp} = 0.2$ to keep graphene sheets stay together[32–34].

## Reporting summary

Further information on research design is available in the Nature Portfolio Reporting Summary linked to this article.

## Data availability

The data generated in this study are provided in the Supplementary Information and Source Data file. Source data are provided with this paper.

## Code availability

The code supporting this study is available from the corresponding author upon request.

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

## Acknowledgements

This work was supported by the financial support from the National Natural Science Foundation of China (No. 52022051, 22035005, 22075165, 52090032, and 52073159 to L.Q.; 22075019 to Y.Z.; 11972349, 11790292 to F.L.; No. 22205129 to B.L.; No. 12302089 to L.Y.); Tsinghua-Foshan Innovation Special Fund (2020THFS0501 to L.Q.), the Strategic Priority Research Program of the Chinese Academy of Sciences (Grant No. XDB22040503 to F.L.).

## Author contributions

L.Q. proposed the research direction and supervised the project. B.L., Y.Z., and L.Q. designed the experiments. B.L. carried out the materials synthesis and all characterizations with help from Y.H., Y.W., F.Z., H.C., and Y.Z. B.L., Y.H., and Y.Z. were involved in data analyzing. B.L., L.Y., and F.L. conducted theoretical simulation. B.L. wrote the manuscript under the advice given from F.Z., Y.Z., F.L., and L.Q. All authors contributed to discussions.

## Competing interests

The authors declare no competing interests.
