## [Peer Review File · Nature Communications]

Evaporate-casting of Curvature Gradient Graphene Superstructures for Ultra-high Strength Structural MaterialsREVIEWER COMMENTS

Reviewer #2 (Remarks to the Author):

This manuscript entitled "Evaporate-casting of Curvature Gradient Graphene Superstructures for Ultra-high Strength Structural Materials" reported a novel processing strategy toward graphene superstructure with a curvature gradient and meticulously design-ability of functional shapes. The content of this paper is detailed. The experiments are well designed and original insights are provided in this paper. It is acceptable for publication in "Nature Communications" after major revision.

As a structural material, the components of the material, interfacial interactions, and the structure are very important to the performance of the material. Although several interesting results were shown in the manuscript, major comments exist as follows:

For the component:

(1) It seems the authors didn't wash the alkali out after the fabrication of cg-G. It is necessary to figure out the main component of the composites or it is inaccurate to name the composite "Graphene Superstructures". Meanwhile, the origin of mechanical performance should be identified. The contribution of each component needs to be clear.

(2) For a more comprehensive understanding of the material's components and properties, more characterizations including FITR, Elemental analysis, and Density (before and after the evaporation casting process) should be included.

For the interface:

(3) Authors should provide additional characterization and discussion on how the graphene structure transitions from porous to compact during the evaporate casting. It would be beneficial to include morphological data of the sample at various stages of evaporation to thoroughly examine the structural changes occurring during this process.

(4) The authors claimed "the hydrogen bond interaction between GO sheets decreased distinctly and electrostatic repulsion increased" (page 19, line 484) but it is lack of evidence. More characterization should be presented to figure out the interfacial interactions between nanosheets.

(5) The interfacial interactions between graphene walls are crucial for their strength, as also indicated by the CG-MD simulation results in Fig. S3. However, the TEM image in Fig. 2f only vaguely depicts these alignments and does not clearly show the interactions between graphene sheets. I recommend employing ultra-high-resolution electron microscopy techniques, such as SEM or FIB-SEM, to more precisely characterize and analyze the interface microstructures within the cg-G.

(6) The bonding interface of the mortise and tenon structure depicted in Fig. 5 e and f should be further magnification and detailed discussion. Providing more detailed SEM images of this area would enhance the understanding of its interface characteristics and excellent mechanical strength.

Process and structure:

(7) Different parameters are suggested to explore. The porous structure is formed in the hydrothermal process and the optimal conditions should be discussed. How to control the gap sizes of layers? Meanwhile, the impact of the gap sizes between the graphene walls in the cg-G before evaporation casting on the structure and mechanical performance of the final product warrants further investigation. I suggest the author investigate how these gaps influence the final properties.

(8) The method of sintering should be explored. The samples obtained from other drying method should be compared to enrich the author's mechanism (The solvent evaporation and interface should be considered together)

Others:

(9) The major concern is the universality. Is this innovative approach feasible for other materials? What are the applicable conditions?

(10) The layout and visualization of Fig. 1 should be further improved to more effectively showcase the novel aspects of the processing strategy and the curvature structures presented in this work. Especially for the process illustration, the current state lacks clarity and is unable to convey the process's uniqueness and casting flexibility.

(11) Regarding the compression tests, the direction in which these were conducted should be specified. If the compressive strength of cg-G was obtained in the in-plane direction, it would be valuable to also include compressive strength data along the out-of-plane direction for a more comprehensive mechanical property comparison and discussion.

Reviewer #3 (Remarks to the Author):

This work focuses on addressing the challenges currently encountered in the fabrication of microstructural materials, successfully develops a novel processing strategy toward graphene superstructure with a curvature gradient, and fabricates a robust structural material with meticulously designed functional shapes. Compared with traditional structural materials, the graphene superstructure fabricated by the dehydration-based evaporate-casting process is tightened via capillary effect, resulting in a compact profile and extraordinary high mechanical strength of up to ~ 70 MPa. This study is very detailed, delving into the preparation and deformation of the graphene superstructure thoroughly, and readers can obtain specific information from the article. This is undoubtedly essential for guiding and advancing subminiature structural materials, addressing challenges faced by current material forming technologies. I have no reservation to recommend this work for publication in Nature Communications after addressing following minor comments.

1. Figure 1 thoroughly illustrates the maturation deformation process of the capsule and the evaporate-casting of curvature gradient graphene. However, the information about the blue dot matrix in Figure 1d is missing. To more clearly obtain information directly from the image, it is recommended to mark the relevant description.
2. Throughout the contraction process, what alterations take place in the geometric structure of normal graphene hydrogel? It is recommended to enhance the article by including configurations of normal graphene hydrogels after contraction, especially highlighting distinctive structures like crosses, dumbbells, butterflies, etc. In addition, what is the C/O ratio determined in graphene hydrogels, and does this ratio significantly influence the preparation of graphene superstructures through curvature gradient structure?
3. In Figure 5a, what distinguishes the representations of the two cube schematics? Are the scale levels consistent, and are there any instances of duplicated information representations? Please confirm. For the mechanical strength test in Figure 5a-d, why the graphene hydrogel is designed as a hollow structure and whether there is a structure-activity relationship? At the same time, is there any difference in mechanical strength of graphene hydrogel between solid and hollow structure?
4. In this study, the deformation of graphene hydrogel derives from the evaporate-casting of curvature gradient structure. In Figure 3, what would occur during the contraction process if the cruciform structure were not disassembled from the concentric circle structure?
5. Is the contraction and deformation process of these graphene superstructures reversible?

RESPONSE TO REVIEWERS' COMMENTS

Reviewer #2 (Remarks to the Author):

This manuscript entitled “Evaporate-casting of Curvature Gradient Graphene Superstructures for Ultra-high Strength Structural Materials” reported a novel processing strategy toward graphene superstructure with a curvature gradient and meticulously design-ability of functional shapes. The content of this paper is detailed. The experiments are well designed and original insights are provided in this paper. It is acceptable for publication in "Nature Communications" after major revision.

Response: Thank you very much for your positive comments. We do appreciate your valuable comments and shall further improve the manuscript as you suggested.

As a structural material, the components of the material, interfacial interactions, and the structure are very important to the performance of the material. Although several interesting results were shown in the manuscript, major comments exist as follows:

For the component:

(1) It seems the authors didn't wash the alkali out after the fabrication of *cg-G*. It is necessary to figure out the main component of the composites or it is inaccurate to name the composite “Graphene Superstructures”. Meanwhile, the origin of mechanical performance should be identified. The contribution of each component needs to be clear. Response: Thank you for your comments. In fact, we have completely removed the residual alkali in this work. The as-formed *cg-G* after hydrothermal process was treated with 1M HCl for 6-8 h to fully react with alkali, followed by thoroughly rinsing process. To avoid the misunderstanding for the preparation procedure of the *cg-G*, more detailed demonstrations have been added in lines 460-462 of the revised manuscript, and highlighted in red for easy reference. Besides, X-ray photoelectron spectroscopy (XPS) and Energy Dispersive Spectroscopy (EDS) analyses of *cg-G* before evaporate-casting have also been conducted and shown in Fig. S6 of the revised Supplementary Information. Both of XPS and EDS revealed only C and O elements in the *cg-G* without any other impurities, confirming the essential characteristics of graphene species. Therefore, the outstanding mechanical properties observed in this work come from solely from the intrinsic characteristics of the graphene material. The corresponding data and analyses have been added and highlighted in red in lines 126-129 of the revised manuscript and Fig. S6 of Supplementary Information.

(2) For a more comprehensive understanding of the material's components and properties, more characterizations including FITR, Elemental analysis, and Density

(before and after the evaporation casting process) should be included.

Response: Thank you for your helpful suggestions. The characterizations of FITR, EDS, and Density for *cg-G* before and after evaporate-casting have been conducted. The detailed analysis and discussion of FITR and EDS analysis have been added in lines 126-129 of the revised manuscript and Fig. S6 of Supplementary Information. The density of *cg-G* before and after evaporate-casting has been added in line 137 of the revised manuscript. All modifications are highlighted in red.

For the interface:

(3) Authors should provide additional characterization and discussion on how the graphene structure transitions from porous to compact during the evaporate casting. It would be beneficial to include morphological data of the sample at various stages of evaporation to thoroughly examine the structural changes occurring during this process.

Response: As suggested, the cuboid-shaped *cg-G* with centrosymmetric curvature gradient structure (corresponding to the one shown in Figure 4c of main text and Fig. S16 of Supplementary Information) was selected as the sample to investigate morphological and structural changes of *cg-G* at various stages of evaporate-casting process.

As depicted in Fig. R1a, it is evident that the appearance of *cg-G* hydrogel gradually changes from a square to a rectangular shape as the contraction process. The observations of microstructures and pore architectures in *cg-G* during the capillary dehydration densification process were conducted and shown in Fig. R1b (here we choose the side view cross-section at position that parallel to the curvature central axis). Initially, the structure of *cg-G* hydrogel shows relatively thin graphene layers (with approximately 20 μm spacing among them) parallel to the central axis of curvature. With the continuous evaporation of water molecules in the hydrogel, the distance between adjacent graphene sheets remains temporarily unchanged. However, wrinkles in the graphene sheets appear and become obvious after 1 hour, due to the influence of capillary forces, which is accompanied by an increase in the thickness of the sheets. Having further evaporated for 3 hours, a significant quantity of water is removed from graphene layers, causing the sheets to wrinkle and further densify. This induces a closer stacking of graphene layers and a decrease in the gap size. After fully removal of water molecules (~ 5 hours later), a compact graphene superstructure with an oriented arrangement of graphene layers is achieved.

It is essential to underscore that during the casting process, the graphene sheets undergo changes in two directions. Along the sheet direction, the graphene sheets become highly curled and rough, whereas perpendicular to the sheet direction, the

wrinkling of the graphene sheets results in an increase in layer thickness and a reduction in interlayer distance. This wrinkling in both directions prevents irreversible stacking of parallel-aligned graphene walls during densification, ensuring the maintenance of the microstructure of the oriented graphene hydrogel after moisture evaporation. The pertinent data along with corresponding analyses have been added and highlighted in red in line 136 of the revised manuscript and Fig. S7 of Supplementary Information.

Fig. R1 a) Digital and b) SEM images of *cg-G* after evaporate-casting in different time (0 h, 1h, 3h, 5h).

(4) The authors claimed “the hydrogen bond interaction between GO sheets decreased distinctly and electrostatic repulsion increased” (page 19, line 484) but it is lack of evidence. More characterization should be presented to figure out the interfacial interactions between nanosheets.

Response: Thank you for your valuable comments. To explore the impact of alkali addition on GO nanosheets, the surface charge and rheological behavior of GO suspension before and after treatment with ammonia were assessed. Upon the addition of ammonia hydroxide (the volume ratio of GO aqueous dispersion to 32% ammonium hydroxide is 60:1), the Zeta potential of the GO dispersion underwent a notable negative shift from an initial -36.2 mV to approximately -47.7 mV (Fig. R2a). This suggests that ammonia can effectively promote the ionization of oxygen-containing functional groups (like carboxyl groups) on the surface of GO, thereby intensifying the electrostatic repulsion between GO sheets. Consequently, this enhancement in electrostatic repulsion dynamically promotes the formation of a liquid crystal phase.

Specifically, the robust electrostatic repulsion between GO sheets significantly enhances the fluidity of the system (Fig. R2b), facilitating the adjustment of GO sheets' orientation in response to external disturbance. This adjustment enables them to migrate to the position of minimal free energy, ultimately resulting in the formation of a long-range ordered liquid crystal phase. Moreover, our previous work (*ACS Nano* 2019, 13, 9161–9170) about the impact of KOH addition on the surface charge and rheological behavior of GO also reveals the similar trend for the enhanced fluidity of the dispersion after adding an appropriate quantity of KOH. The results and discussion have been added and highlighted with red at line 106 of the revised manuscript and Fig. S1 of Supplementary Information.

Fig. R2 a) Zeta potentials, and b) apparent viscosities of GO suspension (3.5 mg mL^{-1}) with/without ammonia hydroxide.

(5) The interfacial interactions between graphene walls are crucial for their strength, as also indicated by the CG-MD simulation results in Fig. S3. However, the TEM image in Fig. 2f only vaguely depicts these alignments and does not clearly show the interactions between graphene sheets. I recommend employing ultra-high-resolution electron microscopy techniques, such as SEM or FIB-SEM, to more precisely characterize and analyze the interface microstructures within the cg-G.

Response: Thank you for your valuable suggestion. In fact, during the spontaneous evaporation-casting process of cg-G, water molecules serve as plasticizer, which enable relative sliding of graphene nanosheets within the walls. In this context, the capillary forces prompt adjacent graphene nanosheets to crumple as water within surface wrinkles evaporates. This would result in the distance between graphene nanosheets decreasing until adjacent layers make complete contact, leading to an increased thickness of graphene sheets. Consequently, a highly dense aligned microstructure with interlocked serrated configurations is formed. To precisely characterize and analyze the interface microstructures within the cg-G, high-resolution SEM characterizations were conducted. As depicted in Fig. R3, the HR-SEM images demonstrate that the oriented behavior of the contracted graphene remains observable, consistent with the findings obtained through TEM analysis of Fig. 2f in revised manuscript. The pertinent data

along with corresponding analyses have been added and highlighted in red in line 136 of the revised manuscript and Fig. S7 of Supplementary Information.

Fig. R3 SEM images under different magnification of *cg-G* after complete evaporate-casting process.

(6) The bonding interface of the mortise and tenon structure depicted in Fig. 5 e and f should be further magnification and detailed discussion. Providing more detailed SEM images of this area would enhance the understanding of its interface characteristics and excellent mechanical strength.

Response: Thank you for your suggestion. We have included SEM images depicting the bonding interface of the mortise and tenon structure, along with magnified views of different regions. As shown in Fig. R4a, it is evident that the two components of the mortise (part a) and tenon structure (part b) are tightly joined together without any gaps. Upon the enlarged SEM view of boundary (Fig. R4b–d), the contact points between the two parts exhibit a rough saw-tooth pattern. It suggests that the mortise (part a) and tenon structure (part b) are getting closer during the spontaneous evaporation-induced contraction of the *cg-G* hydrogel, in which part a happens significant deformation as it wraps around and applies pressure to part b, leading to the formation of a dense connection interface. This resultant tight contact ensures the integrity of the assembled mortise and tenon structure and contributes to its outstanding mechanical properties. The results and discussion have been added and highlighted in red in Fig. S26e of Supplementary Information.

Fig. R4 (a–d) SEM images of the bonding interface of the mortise and tenon structure with different magnification.

Process and structure:

(7) Different parameters are suggested to explore. The porous structure is formed in the hydrothermal process and the optimal conditions should be discussed. How to control the gap sizes of layers? Meanwhile, the impact of the gap sizes between the graphene walls in the cg-G before evaporation casting on the structure and mechanical performance of the final product warrants further investigation. I suggest the author investigate how these gaps influence the final properties.

Response: The preparation process of *cg-G* hydrogels has been thoroughly investigated and optimized.

The gap size of *cg-G* is predominantly governed by the GO concentration during the hydrothermal preparation process. To this end, we selected four different GO concentrations (1.5 mg mL^{-1} , 2.5 mg mL^{-1} , 3.5 mg mL^{-1} , 4 mg mL^{-1}) under identical hydrothermal conditions ($180 \text{ }^\circ\text{C}$, 6h), and characterized the microstructures of the resulting hydrogels. The quantity of alkali increases or decreases proportionally with the amount of GO. At a GO concentration of 1.5 mg mL^{-1} , the resultant graphene hydrogel exhibits an overall disordered state (Fig. R5a), with a pore size of approximately $30\text{-}50 \text{ }\mu\text{m}$. At low GO concentrations, the interaction between GO sheets is weakened, which makes them difficult to form a long-range ordered liquid crystal phase even with the addition of alkali, resulting in an unordered structure of the hydrogel.

Remarkably, uniform and ordered orientation structures can be achieved at concentrations of 2.5 mg mL^{-1} , 3.5 mg mL^{-1} , and 4 mg mL^{-1} (Fig. R5b-d). The disparity

lies in the gap size: at lower concentrations, the spacing is larger (approximately 30-40 μm), whereas with increasing concentration, the interlayer spacing of the orientation structure gradually decreases. At 4 mg mL^{-1} , the gap size is reduced to approximately 10 μm . It is noteworthy that at a concentration of 3.5 mg mL^{-1} , the orientation structure of *cg-G* is remarkably uniform (in this work). The results and discussion have been added and highlighted in red at line 106, 472-475 of the revised manuscript and in Fig. S2 of Supplementary Information.

Fig. R5 SEM images of graphene assemblies with different initial GO concentrations.

To investigate the impact of the initial gap size in the *cg-G* on deformation behavior, we opted cuboid-shaped graphene hydrogels with centrosymmetric curvature gradient structure (corresponding to the one shown in Figure 4c of main text and Fig. S16 of Supplementary Information) at different initial GO concentrations. The initial configuration, dimensions, and microstructure of the cuboid-shaped hydrogels are illustrated in Figs. R6a.

For the initial hydrogels, the ratio of l to w is 1. Following 5 hours of spontaneous water evaporation, all four cuboid-shaped hydrogels underwent contraction and deformation. As shown in Fig. R6b and R6c, it can be observed that the deformation behaviors of the four hydrogels vary significantly. Among them, the hydrogel with an initial GO concentration of 1.5 mg mL^{-1} underwent irregular contraction, with l'/w' approximately 1.1. As the GO concentration increased, the regularity of the *cg-G* significantly improved after evaporate-casting, with the degree of contraction of the w side being noticeably greater than that of the l side, resulting in $l'/w' > 1$. The higher the initial GO concentration, the smaller the l'/w' ratio. The values of l'/w' corresponding to the three GO concentrations are 1.7, 1.4, and 1.2, respectively. The variations in the l'/w' ratios primarily stem from differences in the quantity of graphene

sheets oriented perpendicular to the sheet direction. Higher initial concentrations of GO result in a greater number of graphene sheets within the same volume, leading to increased size and reduced shrinkage upon complete evaporation.

As discussed in the main text, this indicates an anisotropic contraction behavior that could significantly enhance the effectiveness of curvature gradient-induced deformation. Furthermore, the enhancement of deformation behavior diminishes with increasing initial GO concentration. Ultimately, considering the regularity of the deformation process and the degree of deformation behavior, we selected hydrogels with an initial concentration of 3.5 mg mL^{-1} and an initial gap size of $20 \text{ }\mu\text{m}$ as the precursor for studying the evaporate-casting deformation behavior of curvature gradient graphene hydrogels. The results and discussion have been added at lines 249-251 of the revised manuscript and in Fig. S18 of Supplementary Information, highlighted with red.

Fig. R6 a, b) Photographs of cuboid-shaped graphene hydrogels (with different initial GO concentrations) with centrosymmetric microstructure after dehydration of 0 h and 5 h, and c) the corresponding SEM images of dehydration for 5 h. Scale bar: $500 \text{ }\mu\text{m}$. Initial size: $8 \text{ mm} \times 8 \text{ mm} \times 2 \text{ mm}$. The ratios of the two sides along (l) and perpendicular to the wall plane (w) after dehydration (l'/w') are different.

As illustrated in Fig. R7, we conducted a thorough examination of the mechanical properties of graphene hydrogels after water evaporation with various initial GO concentrations. Notably, the mechanical superiority of *cg-G* superstructures over normal graphene assemblies, as discussed in Fig. 2g of the revised main text, is evident.

This discrepancy arises from the influence of capillary forces induced by dehydration during the evaporate-casting process, which drive the deformation of the graphene assembly. At the nanoscale, graphene nanosheets form protruding structures, creating mechanical interlocks. At the assembly level, the layered structure exhibits enhanced orientation and stacking, resulting in extensive contact areas and intensified interlayer interaction. In contrast, disordered normal graphene assemblies only demonstrate mechanical reinforcement at the nanoscale, leading to inferior mechanical properties compared to *cg-G* superstructures.

Additionally, all samples of the other three types of *cg-G* superstructures exhibited exceptional mechanical compression performance. These compression curves consist of two distinct phases: elastic deformation and plastic deformation. The yield point occurs at approximately 8% deformation, while the prolonged plastic deformation phase is attributed to the presence of numerous nanoscale pores within the material. During compression, these pores undergo compression until deformation reaches approximately 50%-60%, ultimately leading to structural collapse and failure. The presence of the plastic phase significantly enhances the material's practical application performance, preventing direct failure under high stress and minimizing damage to instruments or devices. The discussion has been added in line 140 of the revised manuscript and Fig. S8 of Supplementary Information, and highlighted with red.

Fig. R7 Compressive stress-strain curves of the *cg-G* with different initial GO concentrations after evaporate-casting.

(8) The method of sintering should be explored. The samples obtained from other drying method should be compared to enrich the author's mechanism (The solvent evaporation and interface should be considered together)

Response: Thank you for your comments and suggestions. Regarding the drying method employed in this study, we utilized ambient room temperature spontaneous evaporation drying in the laboratory environment. The temperature was maintained at 25 °C, with no significant airflow (such as fans or air conditioning). To minimize the

influence of the casting substrate on the evaporate-casting deformation process, we selected ultra-hydrophobic PTFE board as the deformation substrate. PTFE exhibits a large contact angle ($\sim 170^\circ$), which minimizes the surface tension between the *cg-G* hydrogel and the substrate upon contact, thereby reducing the restriction of the substrate on the evaporate-casting shrinkage process.

To investigate the effects of different drying methods on *cg-G* evaporate-casting behavior, we selected three *cg-G* hydrogel samples with identical internal orientation microstructures. These samples were subjected to drying outdoor, convection oven, and drying in the laboratory environment, respectively. The outdoor drying occurred in Beijing, China, at noon on March 22nd, with a temperature of approximately 23 °C and a northwest wind of 3 levels. The convection oven was set at 35 °C with forced air circulation. Ultra-hydrophobic PTFE was used as the shrinking substrate for all samples.

The experimental results are presented in Fig. R8. After the evaporate-casting process, all three *cg-G* superstructures exhibited almost identical configurations and dimensions. This indicates that factors such as air flow velocity and temperature, which can alter the evaporation rate of water, do not affect the outcome of the evaporate-casting deformation behavior. However, it was observed that the time required for evaporation deformation varied significantly across different environments. In outdoor conditions, higher winds accelerated the evaporation of moisture within the *cg-G* hydrogel, hastening the evaporate-casting process, which was completed in just 2 hours. Furthermore, in the convection oven environment, the combination of higher temperature and faster airflow further accelerated moisture evaporation, completing the shrinkage deformation of *cg-G* in just 1 hour.

These results demonstrate that different drying methods only affect the timing of the deformation process, without impacting the final structure. However, due to the uncontrollable nature of outdoor environmental conditions and the high energy consumption associated with convection ovens, the ambient room temperature spontaneous evaporation method is still preferred for practical experimental purposes, which requires no special treatment and consumes no energy. The discussion has been added and highlighted with red at line 251 of the revised manuscript and in Fig. S19 of Supplementary Information.

Fig. R8 Characterizations of *cg-G* a) before and b) after evaporate-casting with different drying method. Scale bar: 500 μm .

Others:

(9) The major concern is the universality. Is this innovative approach feasible for other materials? What are the applicable conditions?

Response: Thank you for your valuable input. In fact, this evaporate-casting process can also be applicable for other similar hydrogel systems with oriented microstructures. For instance, the oriented MXene hydrogels based on our previous research (*ACS Nano* 2020, 14, 8, 10471–10479) show similar contractive behaviors with graphene superstructures in this work.

Specifically, the preparation of MXene hydrogels followed a meticulously outlined procedure. The $\text{Ti}_3\text{C}_2\text{T}_x$ dispersion (60 mg g^{-1}) was carefully poured into PTFE molds, positioned atop a 0.5 cm thick stainless-steel block in contact with liquid nitrogen. Following the complete freezing of the $\text{Ti}_3\text{C}_2\text{T}_x$ dispersion, the resultant frozen sample was extracted from the PTFE mold and stored in a refrigerator ($T = -20 \text{ }^\circ\text{C}$) for a duration of 4 hours. Subsequently, the frozen gel underwent a thawing process in 5 M HCl solution at room temperature, with gentle agitation, spanning a timeframe of 6 hours. The solution was then subjected to multiple replacements with deionized water until achieving neutrality. The subsequent dialysis process culminated in the preparation of MXene hydrogel characterized by a long-range oriented microstructure.

Fig. R9 illustrates the ensuing steps wherein the obtained MXene hydrogel was

meticulously sliced into dimensions of $8\text{ mm} \times 8\text{ mm} \times 2\text{ mm}$ and positioned on a PTFE substrate, allowing for spontaneous evaporation of water under ambient conditions. Observations from Fig. R9a elucidate a significant reduction in dimensions perpendicular to the layer direction as water evaporated, indicating a marked decrease in gap size, whereas changes along the layer direction remained relatively minor. After 5 hours, the complete evaporation of water molecules within the MXene hydrogel culminated in its final evaporative shaping. Analogous to curvature gradient graphene hydrogels, the resultant MXene assembly manifested a regular rectangular structure, underscoring the controllable nature of MXene hydrogel shrinkage.

Further analysis through SEM images (Fig. R9b) revealed a distinct long-range ordered structure within the MXene hydrogel prior to shrinkage, characterized by an interlayer spacing of approximately $50\text{ }\mu\text{m}$. With ongoing water evaporation, the sheets of MXene hydrogel exhibited wrinkling, leading to the convergence of adjacent layers and a subsequent decrease in interlayer spacing until its eventual disappearance (Fig. R9c). This behavior mirrors the shrinkage exhibited by curvature gradient graphene hydrogels. Nonetheless, the degree of wrinkling observed in MXene hydrogel sheets was comparatively lower. Consequently, the degree of shrinkage along the layer direction within MXene hydrogel was diminished, with the l'/w' ratio after deformation reaching as high as ~ 2 , significantly surpassing that of curvature gradient graphene superstructures.

These findings underscore the evaporative-induced deformation behavior shared between MXene hydrogels and *cg-G* hydrogels, thereby validating the proposed method's universality. Similarly, by replicating analogous oriented or curvature gradient structures, it is hypothesized that other materials such as graphene/MXene composite hydrogels or polymer composite hydrogels can likewise exhibit comparable evaporative deformation behavior. The discussion has been added at line 361 and 362 of the revised manuscript and in Fig. S30 of Supplementary Information, highlighted with red.

Fig. R9 a) Optical images of vertically oriented MXene hydrogel with evaporation time ranging from 0 to 5 h. SEM images of the MXene hydrogel b) before and c) after water evaporation. d) EDS mappings of Ti, C, F, and O elements. Scale bar: 50 μm .

(10) The layout and visualization of Fig. 1 should be further improved to more effectively showcase the novel aspects of the processing strategy and the curvature structures presented in this work. Especially for the process illustration, the current state lacks clarity and is unable to convey the process's uniqueness and casting flexibility.

Response: As suggested, we have revised the schematic diagram of the evaporate-casting process of *cg-G* in Fig. 1 of the revised main text. The modified structure clearly illustrates the macroscopic and microscopic changes in the curvature gradient structure before and after water evaporation. Additionally, we have included the revised Fig. 1 in the subsequent figure (Fig. R10) for reference.

Fig. R10 The revised Fig. 1 in main text.

(11) Regarding the compression tests, the direction in which these were conducted should be specified. If the compressive strength of cg-G was obtained in the in-plane direction, it would be valuable to also include compressive strength data along the out-of-plane direction for a more comprehensive mechanical property comparison and discussion.

Response: Thank you for your comment and suggestion. Fig. R11 provides depicts the stress-strain curves of cg-G after deformation at two different directions. As evident from the graph, cg-G demonstrates notably high strength (~100 MPa) in two perpendicular stress directions. However, there are differences in the extent of deformation at the yield point. The difference lies in the fact that there is a greater Young's modulus along the direction of the sheet, while there is greater elastic and plastic deformation perpendicular to the direction of the sheet. This indicates that the mechanical properties are favorable in both directions, and in practical applications, the loading surface can be chosen according to specific needs. The results and discussion have been added at line 140 of the revised manuscript and in Fig. S8 of Supplementary Information, highlighted with red.

Fig. R11 Compressive stress-strain curves of the *cg-G* after evaporate-casting under different compression directions.

Reviewer 3

This work focuses on addressing the challenges currently encountered in the fabrication of microstructural materials, successfully develops a novel processing strategy toward graphene superstructure with a curvature gradient, and fabricates a robust structural material with meticulously designed functional shapes. Compared with traditional structural materials, the graphene superstructure fabricated by the dehydration-based evaporate-casting process is tightened *via* capillary effect, resulting in a compact profile and extraordinary high mechanical strength of up to ~ 70 MPa. This study is very detailed, delving into the preparation and deformation of the graphene superstructure thoroughly, and readers can obtain specific information from the article. This is undoubtedly essential for guiding and advancing subminiature structural materials, addressing challenges faced by current material forming technologies. I strongly recommend publishing this work in Nature Communications after minor revision.

Response: We do appreciate your positive and valuable comments, and shall further improve the manuscript according to your suggestion.

1. Figure 1 thoroughly illustrates the maturation deformation process of the capsule and the evaporate-casting of curvature gradient graphene. However, the information about the blue dot matrix in Figure 1d is missing. To more clearly obtain information directly from the image, it is recommended to mark the relevant description.

Response: Thank you for your valuable suggestion. We have revised the schematic diagram of the evaporate-casting process of *cg-G* in Fig. 1 of the revised main text. The modified structure clearly illustrates the macroscopic and microscopic changes in the curvature gradient structure before and after water evaporation.

2. Throughout the contraction process, what alterations take place in the geometric structure of normal graphene hydrogel? It is recommended to enhance the article by including configurations of normal graphene hydrogels after contraction, especially highlighting distinctive structures like crosses, dumbbells, butterflies, etc. In addition, what is the C/O ratio determined in graphene hydrogels, and does this ratio significantly influence the preparation of graphene superstructures through curvature gradient structure?

Response: For the geometric structure of normal graphene hydrogels, the contraction process entails a reduction in volume with irregular altering in the shape. This is attributed to the irregular nature of the internal structure of normal graphene hydrogels. During the shrinkage process, the local asymmetric microstructure causes internal stress, which leads to uncontrollable deformation. In the paper, we have incorporated the demonstrations of normal graphene hydrogels with different geometric structures

after contraction (Fig. R12). The results and discussion have been added and highlighted with red in Fig. S14 of Supplementary Information.

Analysis *via* X-ray photoelectron spectroscopy indicates an oxygen composition of approximately 20% and C/O ratio of ~ 4 within the graphene hydrogel. The quantity of oxygen functional groups plays a significant role in influencing the interaction between water molecules and graphene walls. As the number of oxygen functional groups increases, the hydrogen bonding interaction between water molecules and graphene walls strengthens, reducing the ease of the dehydration process. In essence, the contraction and deformation of *cg-G* take more time when there are more oxygen functional groups.

Fig. R12 Digital and SEM images of normal graphene hydrogels with different geometric structures (a) before and (b) after contraction.

3. In Figure 5a, what distinguishes the representations of the two cube schematics? Are the scale levels consistent, and are there any instances of duplicated information representations? Please confirm. For the mechanical strength test in Figure 5a-d, why the graphene hydrogel is designed as a hollow structure and whether there is a structure-activity relationship? At the same time, is there any difference in mechanical strength of graphene hydrogel between solid and hollow structure?

Response: Thank you for your comment. The two cube schematics in Fig. 5a are essentially identical. However, the upper diagram illustrates the external configuration of the hollow square structure, while the lower diagram depicts the microstructure of the hollow square structure. Both diagrams belong to the same dimensional scale. To prevent readers from perceiving similar information as redundant, we have adjusted the perspective of the schematic diagram below to better correspond with the physical image. The relevant modifications have been implemented in Fig. 5a.

In the context of structural materials, lightweight and high strength are pivotal physical attributes. The design of hollow structures aims to reduce the volume density

of the material while maintaining its strength, thereby improving the specific strength of the material and making it more suitable for aerospace applications. In our experiments, we observed that incorporating cylindrical hollow structures within concentric circular arrangements does not compromise the compressive strength of the material. This is because, in compression experiments, the remarkable strength of the graphene superstructure primarily stems from the oriented contraction of graphene layers, and the interlocking interactions between layers. The hollow structure does not affect these microscopic interactions. Additionally, we conducted compression tests on solid graphene blocks, and when subjected to the pressure of a car passing over them, the solid and dense graphene blocks did not undergo destructive deformation (Fig. 2a). Furthermore, we conducted a comparison of the stress-strain curves between hollow and solid *cg-G* after evaporate-casting (Fig. 2b). It is evident that the stress-strain curve of the hollow one exhibits a similar shape and magnitude to that of the solid one. In essence, the mechanical performance of the dense graphene block is not adversely affected by the presence of the hollow structure. Considering the effective reduction in mass achieved by the hollow structure in the *cg-G* after evaporate-casting, it emerges as a more favorable configuration for structural materials. The discussion has been added in Fig. S25 of Supplementary Information, highlighted with red.

Fig. R13 a) Photograph of evaporate-casted solid *cg-Gs* after being run over by car. After being run over by a car, no damage can be found on the five micro-fixators of *cg-G* after evaporate-casting. b) Compressive stress-strain curves of the solid and hollow *cg-Gs* after evaporate-casting.

4. In this study, the deformation of graphene hydrogel derives from the evaporate-casting of curvature gradient structure. In Figure 3, what would occur during the contraction process if the cruciform structure were not disassembled from the concentric circle structure?

Response: Thank you for your comment. In Figure 3, the cruciform structure which was not disassembled from the concentric circle structure will undergo contraction without

any additional deformation.

As outlined in our paper, the deformation of graphene superstructures derives from the evaporate-casting of curvature gradient structure. If the cruciform structure remains integrated with the concentric circular structure, each edge/part of the cruciform exhibits a centrally symmetrical arrangement within the concentric circles. Throughout the contraction deformation process, both structures maintain the same size but in opposite direction of the configuration forces, resulting in an overall absence of deformation. In essence, the concentric circular structure of graphene hydrogel, post-dehydration, contracts into smaller concentric circles, and any internal structure undergoes only dimensional changes without altering its shape compared to the initial hydrogel. We have already delved into this phenomenon in Fig. S12 of revised Supplementary Information. As shown in Fig. S12, the *cg-G* of $d = 0$ mm (blue) mostly consists of arcs with small radii and gaps between them, while the *cg-G* of $d = 7$ mm (green) mostly consists of arcs with large radii. In an ideal coaxial circular arc model, the assembly tends to contract uniformly. Various curvature structures within assemblies demonstrate consistent lateral dimensions following contraction deformation.

5. Is the contraction and deformation process of these graphene superstructures reversible?

Response: The contraction and deformation process of graphene superstructures are irreversible due to the robust π - π interaction between graphene layers. Throughout the hydrogel shrinkage process, the loss of water leads to a tight bonding of adjacent graphene layers through this formidable force. The re-infiltration of water molecules does not reverse the contraction of the graphene layers that have already occurred.

REVIEWERS' COMMENTS

Reviewer #2 (Remarks to the Author):

The authors have addressed the questions very well with supporting experimental data. It is good for publication.

Reviewer #3 (Remarks to the Author):

This manuscript seems now ready for publication.

RESPONSE TO REVIEWERS' COMMENTS

Reviewer #2 (Remarks to the Author):

The authors have addressed the questions very well with supporting experimental data. It is good for publication.

Re. Thank you very much for your positive comments and your time on our manuscript.

Reviewer #3 (Remarks to the Author):

This manuscript seems now ready for publication.

Re. Thank you very much for your positive recommendation.